# Mouse *Slfn8* and *Slfn9* genes complement human cells lacking *SLFN11* during the replication stress response

Erin Alvi[1,3], Ayako L. Mochizuki[1,4], Yoko Katsuki [1,5], Minori Ogawa[1], Fei Qi[1], Yusuke Okamoto[1,6], Minoru Takata[1,2] & Anfeng Mu [1,2✉]

The *Schlafen (SLFN)11* gene has been implicated in various biological processes such as suppression of HIV replication, replication stress response, and sensitization of cancer cells to chemotherapy. Due to the rapid diversification of the *SLFN* family members, it remains uncertain whether a direct ortholog of human *SLFN11* exists in mice. Here we show that mSLFN8/9 and hSLFN11 were rapidly recruited to microlaser-irradiated DNA damage tracks. Furthermore, *Slfn8/9* expression could complement *SLFN11* loss in human *SLFN11*$^{-/-}$ cells, and as a result, reduced the growth rate to wild-type levels and partially restored sensitivity to DNA-damaging agents. In addition, both *Slfn8/9* and *SLFN11* expression accelerated stalled fork degradation and decreased RPA and RAD51 foci numbers after DNA damage. Based on these results, we propose that mouse *Slfn8* and *Slfn9* genes may share an orthologous function with human *SLFN11*. This notion may facilitate understanding of *SLFN11*'s biological role through in vivo studies via mouse modeling.

[1] Laboratory of DNA Damage Signaling, Department of Late Effects Studies, Radiation Biology Center, Graduate School of Biostudies, Kyoto University, Kyoto, Japan. [2] Multilayer Network Research Unit, Research Coordination Alliance, Kyoto University, Kyoto, Japan. [3] Present address: Laboratory of Biochemical Cell Dynamics, Institute for Integrated Cell-Material Sciences (WPI-iCeMS), Graduate School of Biostudies, Kyoto University, Kyoto, Japan. [4] Present address: CiRA Foundation, Kyoto, Japan. [5] Present address: Department of Cellular Biochemistry, Graduate School of Pharmaceutical Sciences, Kyushu University, Fukuoka, Japan. [6] Present address: Lunenfeld-Tanenbaum Research Institute, Mount Sinai Hospital, Toronto, ON, Canada. ✉email: mu.anfeng.7x@kyoto-u.ac.jp

The *Schlafen (SLFN)* gene family members have been implicated in a range of biological processes including T-cell development, viral immunity, replication stress, and cell fate decisions following cancer chemotherapy[1–3]. The *SLFN* genes are mostly mammalian specific, and classified into the subgroups I, II, or III depending on the domain, structure and size of the protein[1,2]. SLFNs all share the Schlafen core domain including the N-terminal AAA_4 domain and the SLFN box, while subgroup II and III members additionally contain the SWAVDL domain. Only subgroup III proteins harbor the DNA/RNA helicase domain at their C-terminus, which is connected with the Schlafen core by the Linker domain (containing the SWAVDL domain), making them the longest among SLFNs. At present the function of these domain features have been poorly defined. The best-studied is perhaps the N-terminal AAA-4 domain whose structure was elucidated in rat SLFN13[4] or human SLFN5[5]. The domain participates in the control of translation by targeting tRNA/rRNA as an endonuclease and exerts anti-HIV activity in hSLFN13[4] as well as in hSLFN11[6]. But not all of the SLFN members cleave tRNA[5], and hSLFN5 suppresses HIV transcription via an epigenetic mechanism[7]. In addition, a recent study comprehensively described the structure of SLFN11, elucidating critical aspects such as dimerization, binding sites to tRNA and single-strand DNA (ssDNA). These findings offer valuable insights into the functional mechanisms of SLFN11[8]. However, the core biological function that SLFN members may exert with these domains and how they are regulated remains unclear.

Currently, it is understood that the expression of *SLFN11* facilitates cell death after DNA damaging cancer chemotherapies, enhancing clinical efficacy[9,10] or preventing relapse[11]. *SLFN11*, a member of subgroup III, now has been the focus of an increasing research interest based on its potential clinical utility as a biomarker to predict therapeutic responses[12,13]. Importantly, human clinical tumors often lose expression of *SLFN11* during carcinogenesis and following chemotherapeutic treatments because of epigenetic silencing, and common human cancer cell lines often lack its expression[10]. *SLFN11* is therefore suggested to be a potential tumor suppressor[14]. Mechanistically, *SLFN11* suppresses DNA repair activity due to homologous recombination and affects checkpoint maintenance[15], blocks replication fork progression[16], controls transcription of the immediate early genes[17], promotes the degradation of the replication factor CDT1[14], and suppresses the unfolded protein response[18].

Our previous study indicated that *SLFN11* accelerates stalled fork degradation upon replication stress or after DNA damage due to nucleases like MRE11 or DNA2 by preventing recruitment of the fork protector RAD51[19]. We confirmed this role of *SLFN11* in cells with the genome instability disorder Fanconi anemia (FA), in which the compromised fork stability is ameliorated by the depletion of *SLFN11*, but also in the wild-type setting. Thus we have proposed that this fork instability could be one mechanism for the enhanced DNA damage sensitivity[19]. Many of these proposed mechanisms as mentioned above were generally shown to depend on the helicase domain. Conversely, it was reported that hSLFN11 downregulates protein levels of ATR kinase, which is critical for cellular response to DNA damage and replication stress, during DNA damage response depending on its RNase activities, leading to decreased viability following DNA damage[20].

Given the multitude of proposed mechanisms, there still seems to be no unified understanding of how *SLFN11* promotes cell death after DNA damage. Moreover, there is insufficient explanation regarding why *SLFNs* have been rapidly evolving and have diverged together with the other immune-related genes in mice[21] and perhaps in the other species. Hematopoietic and immune cells may have higher expression levels of SLFNs which could be enhanced by interferon[22], although *SLFNs* are ubiquitously expressed. In addition, the cross-species relationship between *SLFNs* is often not very clear. For example, mice have ten *Slfns* (*Slfn1/1L/2/3/4/5/8/9/10/14*), while humans express only six (*SLFN5/11/12/12L/13/14*)[1]. Humans do not have the counterparts of the subgroup I *SLFNs* in mice, and the interspecies relationships among subgroup III *SLFNs* are not immediately apparent except for orthologous *SLFN5-Slfn5* and *SLFN14-Slfn14* pairs. The remaining members of subgroup III *SLFNs* in humans are *SLFN11* and *SLFN13*, while in mice, there are *Slfn8* and *Slfn9* (*Slfn10* being a pseudogene).

In this study, we wished to obtain insights about *SLFN11/13* vs *Slfn8/9* cross-species relationships. We reasoned that such knowledge would facilitate planning of mice models to study important aspects of *SLFN* biology including potential function in carcinogenesis, cancer chemotherapy or blood disorders like Fanconi anemia. Here we show that mouse SLFN8 and SLFN9 behave similarly to hSLFN11 in both human (*SLFN11*$^{-/-}$) and mouse (*Slfn8/9/10*$^{-/-}$) knockout cell lines. Consistent with our recent proposal that *SLFN11* may enhance DNA damage sensitivity by accelerating degradation of the stalled replication forks, expression of *Slfn8* or *Slfn9* in *SLFN11*$^{-/-}$ cells destabilized the nascent DNA track following HU treatment. These results may support the notion that mouse *Slfn8* and *Slfn9* genes share the orthologous function of the *SLFN11* gene.

## Results

### Sequence conservation among subgroup III SLFNs in humans and mice

Given the implication of *SLFN11* in genome stability and cancer, we sought to identify the mouse ortholog of human *SLFN11*. In humans and mice, the paralogous *SLFN* genes cluster within a syntenic region on either human chromosome 17 or mouse chromosome 11 (Fig. 1a). In both species, the region is flanked by the genes *PEX12/Pex12* and *UNC45B/Unc45b*. The orthologous *SLFN5/Slfn5* and *SLFN14/Slfn14* genes are similarly located at the far left and right ends, respectively, within the locus of both species. The other subgroup III *SLFN* genes (i.e., human *SLFN11/13* and mouse *Slfn8/9/10*) seem to be located at corresponding positions, however, their orthologous inter-relationship is not immediately apparent. The sequence alignment analyses using the MAFFT program could not reveal the cross-species correspondence between them (Fig. 1b). In mice, *Slfn8* and *Slfn9* protein sequences are highly similar (86.6%). In addition, *Slfn10* is highly similar to *Slfn8* and *Slfn9*, but is a known pseudogene, and so it was not included in this analysis. Human *SLFN11* and *SLFN13* are also highly similar (77.5%). Human SLFN11 and mouse SLFN8 or SLFN9 were 60.2% or 61.5% identical, respectively, whereas human SLFN13 had a slightly higher homology with mSLFN8 and mSLFN9 (63.5% and 63.9% identity). In contrast, human SLFN5 or SLFN14 have the highest homology with the mice orthologs compared to the other paralogs, supporting their orthologous relationship. These results may be consistent with the previous suggestion that *Slfn8* and *Slfn9* are the orthologs of human *SLFN13*, and thus implying there is no *SLFN11* ortholog in mice[4,5]. However, this notion seems to be at odds with the fact that *SLFN13* lacks the nuclear localization signal (NLS). It is localized (perhaps mostly) in the cytoplasm[4], whereas the other subgroup III SLFNs including mSLFN8 and SLFN9 have the NLS and are primarily localized in the nucleus[23]. We also note one possible discrepancy in the tissue expression pattern between *SLFN13* and *Slfn8/9*. *SLFN11* has much greater expression than *SLFN13* in hematopoietic progenitors. Likewise, *Slfn8*, and *Slfn9* in a lesser degree, are well expressed in mouse hematopoietic cells (https://gexc.riken.jp/, Supplementary Fig. 1)[24].

Interestingly, human SLFN11 and SLFN13 or mouse SLFN8 and SLFN9 maintain overall domain structures, such as the N-terminal Schlafen core domain, the Linker domain, and the

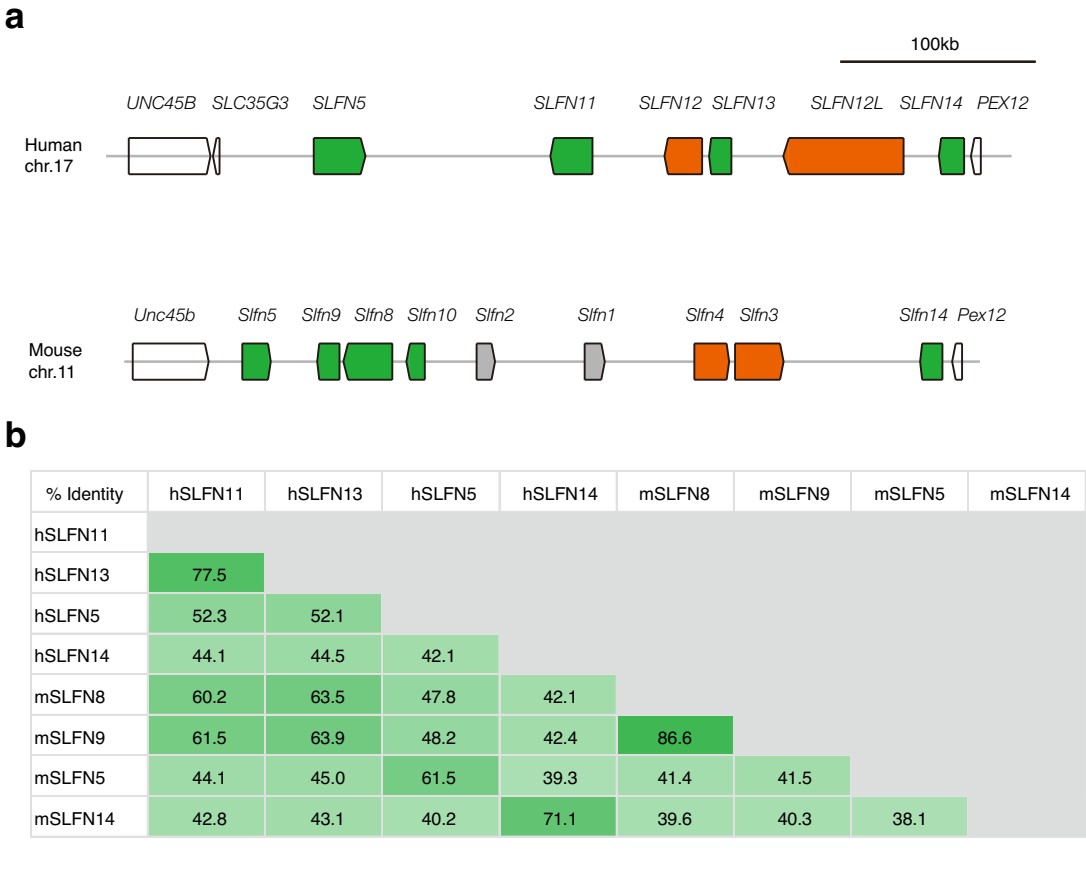

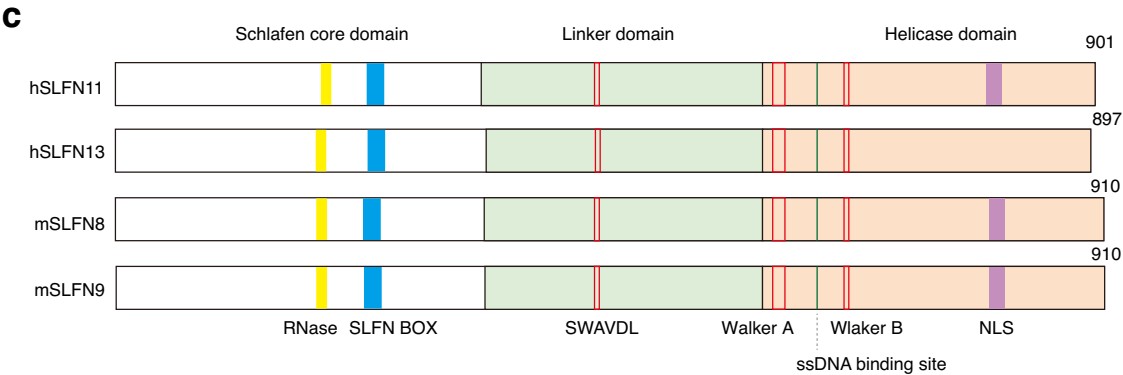

**Fig. 1 Conservation and relationship between human and mouse *SLFNs*. a** Chromosomal location of *SLFN* genes on the mouse and human chromosomes adaptd from the NCBI website (https://www.ncbi.nlm.nih.gov/gene/). The genomic regions shown are chr17:35145202–35579685 (hg38) and chr11:82897551–83307521 (mm10). Gray, subgroup I. Orange, subgroup II. Green, subgroup III. **b** Percentage of the amino acid identity between indicated human and mouse SLFNs. **c** Conserved regions of human SLFN11/13 and mouse SLFN8/9.

C-terminal helicase domain. Also notable, the critical amino acid residues in the ribonuclease domain (especially at the catalytic Glu and Asp residues)[4], ssDNA binding site[8] and helicase domain (Walker type A and B motifs) are also well conserved in all of these *SLFNs* (Fig. 1c and Supplementary Fig. 2). Indeed, biochemical assays showed that human SLFN11, rat SLFN13 and mouse SLFN8 have similar endoribonuclease activity[4,5,8]. Residues in the two dimerization interfaces[8] are also well conserved (Supplementary Fig. 2). We looked at the Alfafold 2 prediction database (https://alphafold.ebi.ac.uk/)[25,26]. Consistent with their highly conserved primary structure, the functional domains of mouse SLFN8 and SLFN9 are predicted to fold in a similar manner to the reported structures of SLFN11 and SLFN13. These considerations suggest that we need functional

analyses in cells to interrogate the cross-species relationship between *SLFNs* in humans and mice.

**hSLFN11, mSLFN8 and mSLFN9, but not hSLFN5, hSLFN13 and mSLFN2, are recruited to DNA damage sites**. The subgroup III *SLFNs* are structurally similar to each other, although previous studies indicated their distinct functions. For example, among human *SLFNs*, only *SLFN11* has been implicated in affecting cell fate decisions after cancer chemotherapy[10], suggesting that only *SLFN11* participates in the DNA damage response, though SLFN5 has recently been implicated in the 53BP1 topological regulation and non-homologous end joining[27]. Indeed SLFN11 had been shown to accumulate at the

laser-induced DNA damage site 40 min later or DNA damage-induced foci[15]. To functionally evaluate the subgroup III *SLFNs* on DNA damage response, we tested their accumulation at DNA damage sites. We first transiently expressed human SLFN11 tagged with GFP at its C-terminus in the osteosarcoma cell line U2OS, which does not express SLFN11, treated with the sensitizer Hoechst33342, and applied 405 nm laser irradiation. The expression plasmids were verified by 293 T cell transfection and western blotting (Supplementary Fig. 3a). We chased the kinetics of relative fluorescence intensity to the pre-irradiated value in the region of interest. We could detect modest but rapid (within less than a minute) accumulation of hSLFN11-GFP at the laser stripes (Fig. 2a), after the initial dip because of photobleaching. Prior studies indicated that SLFN11 interacts with RPA and is distributed within the nucleus in a manner depending on RPA[15]. However, we observed a only partial reduction in hSLFN11-GFP recruitment in cells that underwent RPA knockdown (Fig. 2a and Supplementary Fig. 3b). In contrast, the SLFN11 mutant deficient in ssDNA binding (K652D) hardly accumulated on the micro-laser stripe (Fig. 2b and Supplementary Fig. 3c)[8]. These results indicate the essential role of ssDNA binding in SLFN11 recruitment immediately after DNA damage, and the RPA interaction contributes only partially. The SLFN11 deleted with C-terminal 161 amino acids, known to interact with RPA[15], was not recruited to the DNA damage sites either (Fig. 2b). However, this region contains a NLS, and we observed that the proteins tended to distribute outside of the nucleus, though we have set the region of interest inside nuclei. To more clearly define the role of RPA in SLFN11 recruitment, a SLFN11 missense mutant abrogating RPA binding would be required. Interestingly, RPA1-GFP also rapidly accumulated on the laser stripes with similar kinetics (Fig. 2c).

We then investigated whether the other subgroup III SLFNs (human SLFN5/13 and mouse Slfn8/9, all were tagged with GFP at their C-terminus) are similarly mobilized during DNA damage response. We also chose mSLFN2, one of Group I SLFNs, as a negative control (Supplementary Fig. 3d)[28]. Laser irradiation induced the recruitment of GFP-tagged mSLFN8 or mSLFN9 to the laser tracks, while mSLFN2, hSLFN13, or hSLFN5 were not accumulated (Fig. 2d, e). hSLFN13-GFP was expressed at lower levels and was often distributed in the cytoplasm, consistent with the lack of NLS. We examined only cells expressing nuclear hSLFN13-GFP. It is also interesting to note that hSLFN5-GFP or hSLFN13-GFP often displayed small punctate or speckled spots within the nucleus. mSLFN8-GFP localized in irregular shaped subnuclear bodies. The last observation seems consistent with a previous report that suggested a linkage of mSLFN8 with transcription[23], however, it is currently unclear what is the molecular basis of these distributions. Collectively, These results indicated that mSLFN8 and mSLFN9, but not hSLFN5 or hSLFN13, behave similarly to hSLFN11 immediately after DNA damage.

***Slfn8* and *slfn9* expression reduces the growth rate in human and mouse cells.** The above data prompted us to investigate whether the expression of mouse *Slfn8* or *Slfn9* can restore the phenotype caused by the loss of human *SLFN11*. We infected HAP1 *SLFN11*−/− cells with the doxycycline (DOX) inducible lentivirus vectors encoding GFP-tagged mSLFN8, SLFN9, or hSLFN11, and hygromycin selection was applied. We confirmed the expression of the constructs in each of these selected cell lines after DOX treatment by western blotting and microscopic observation (Fig. 3a). First, we measured the growth of HAP1 *SLFN11*−/− cells with DOX-induced expression of mouse and human SLFNs compared to Wild-type (WT) HAP1 cells, since it has been reported that HAP1 *SLFN11*−/− cells have an increased growth rate

compared to WT cells[19]. We confirmed that the HAP1 *SLFN11*−/− cells grew faster than the WT cells (Fig. 3b), and *Slfn8 or Slfn9* expression lowered the growth rate in *SLFN11*−/− cells in a similar manner to *SLFN11* expression (Fig. 3b).

Notably, either one of the mouse *Slfns* was sufficient to reduce the growth rate of *SLFN11*−/− cells, suggesting that mouse *Slfn8* and *Slfn9* can function redundantly. Accordingly, we decided to make a *Slfn8/9* double knock-out mouse cell line to complement the experiments in human HAP1 cells. We chose the mouse pro-B cell line Ba/F3, which is well characterized and has been used for genome editing experiments[29]. The knockout vector was designed to delete large parts of both *Slfn8* and *Slfn9* genes simultaneously. Because *Slfn8* and *Slfn9* genes and *Slfn10* pseudogene are highly homologous, our CRISPR-Cas9 for cleaving *Slfn8* or *Slfn9* is not very specific, and likely cut the *Slfn10* pseudogene as well (Supplementary Fig. 4a). WT Ba/F3 cells were simultaneously introduced with the targeting vector and two CRISPR vectors and selected with puromycin. We isolated two clones that deleted *Slfn8/9/10* genes at once with this strategy (Supplementary Fig. 4b, c). We observed that the *Slfn8/9/10*−/− cells also had a higher growth rate than WT BaF/3 (Fig. 3c), suggesting simultaneous inactivation of *Slfn8* and *Slfn9* in mouse cells has the same impact on growth as *SLFN11* loss in human cells.

**Sensitivity to DNA-damaging agents is restored with the expression of *SLFNs*.** *SLFN11*−/− cells are resistant to DNA-damaging agents such as cisplatin (CDDP) or replication stress induced by hydroxyurea (HU) (Fig. 3d, e)[19]. CDDP creates DNA adducts, including ICLs, in turn preventing transcription as well as replication, while HU causes fork stalling due to depletion of dNTPs. We observed that HAP1 *SLFN11*−/− cells expressed *SLFN5*, similarly to WT HAP1, but *SLFN13* was not expressed in either of them (Supplementary Fig. 5). We tested if the expression of mouse *Slfn8/9* would restore DNA-damage sensitivity in HAP1 *SLFN11*−/− cells. DOX-induced mouse *Slfn8* or *Slfn9* expression in human *SLFN11*−/− cells using lentivirus partially restored sensitivity to CDDP and HU, to the same level as when *SLFN11* was expressed (Fig. 3d, e). We also generated cells with DOX-inducible expression of GFP-tagged mSLFN2 or hSLFN11-K652D in HAP1 *SLFN11*−/− background (Supplementary Fig. 6a, b). Expression of mSLFN2 did not clearly enhance HU sensitivity in HAP1 *SLFN11*−/− cells (Supplementary Fig. 6c). However, the expression of hSLFN11-K652D still displayed a mild HU sensitivity, indicating that not all SLFN11 function depends on ssDNA binding (Supplementary Fig. 6d).

In mouse Ba/F3 cells, *Slfn8/9/10*−/− cells were more tolerant to HU treatment, while they displayed only slight CDDP sensitivity (Fig. 3f–h). We suppose this could be due to possible low expression of *Slfn8/9* in Ba/F3, or because the role of *SLFNs* is primarily in the replication stress response (induced by HU) rather than in DNA repair/tolerance (these activities handle CDDP damage), or both. To confirm that *SLFNs* can also complement *Slfn8/9* loss in mouse cells, we tested the sensitivity to HU in Ba/F3 *Slfn8/9/10*−/− cells after the expression of *SLFNs* (Fig. 3f, g). Because the lentivirus-mediated transduction into Ba/F3 cells was unsuccessful, we utilized transient plasmid-based expression instead. In line with the previous results, the expression of *SLFN11* restored the same level of sensitivity as the expression of the mouse *Slfn8/9* in mouse *SLFN8/9/10*−/− cells, again suggesting that the hSLFN11 and mSLFN8/9 proteins can function in the same way in both mouse and human cells (Fig. 3g).

***SLFNs* restore replication fork degradation after HU treatment.** Previous studies have revealed that stabilized replication forks

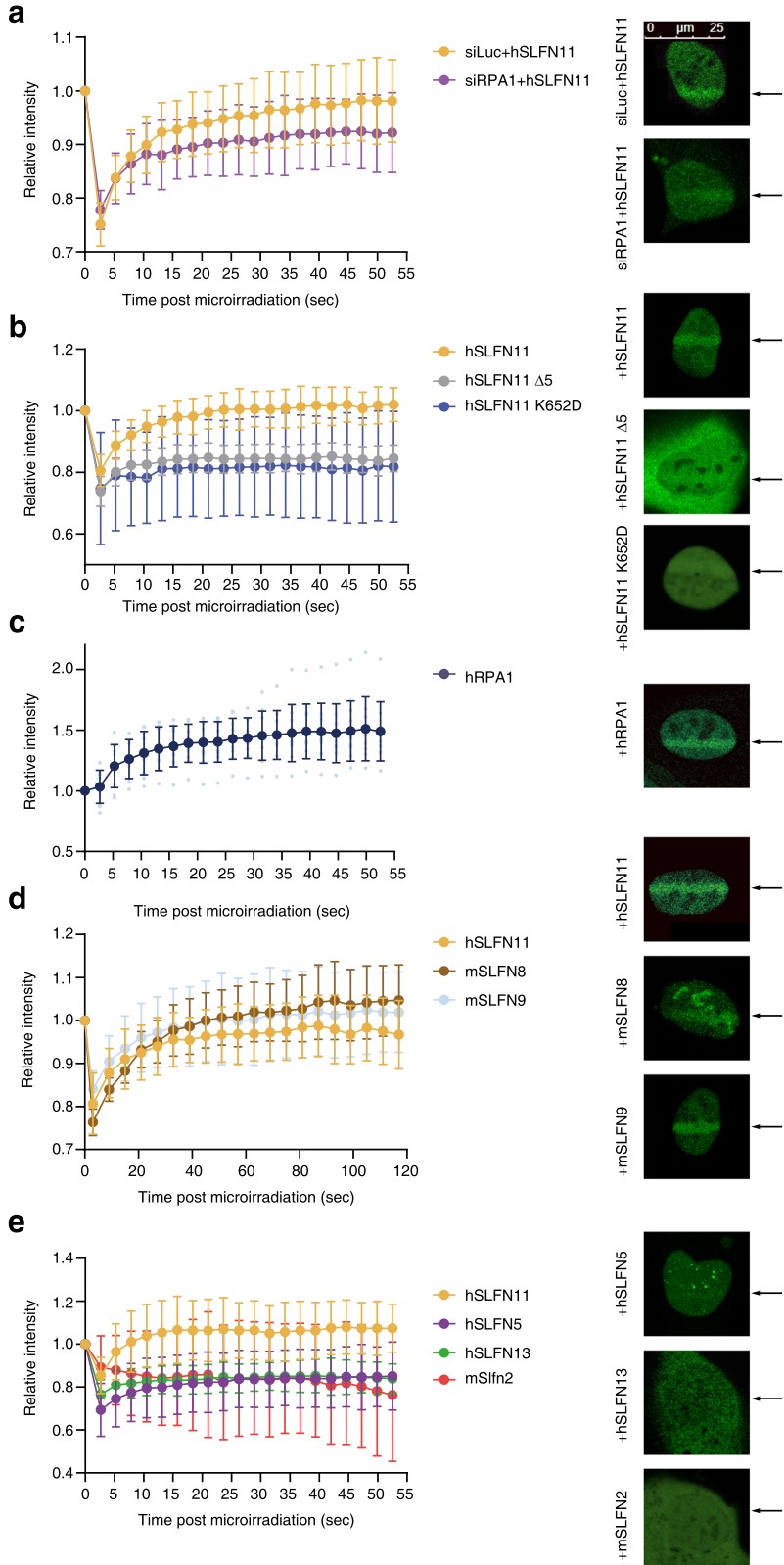

**Fig. 2 Recruitment of GFP-tagged SLNF proteins to 405 nm microlaser-induced DNA damage sites. a** Kinetics of hSLFN11 accumulation to the laser track in U2OS cells 48 h after siLuc and siRPA transfection. The Y-axis represents the fluorescence intensity ratio relative to the pre-irradiated value within region of interest. Kinetics of **b** hSLFN11 Δ5, K652D, **c** RPA1, **d** mSLFN8/9, or **e** mSLFN2, hSLFN13 and hSLFN5 accumulation in U2OS cells were similarly examined. Δ5, *SLFN11* truncation mutant lacking the C-terminal RPA interacting region. Mean ± SD of more than 10 irradiated cells is shown. The right panel showed representative images of the recruitment of SLFNs to DNA damage sites in U2OS cells following laser irradiation. Expression levels of hSLFN13-GFP were low and the image was digitally enhanced. Arrows indicate position of laser tracks.

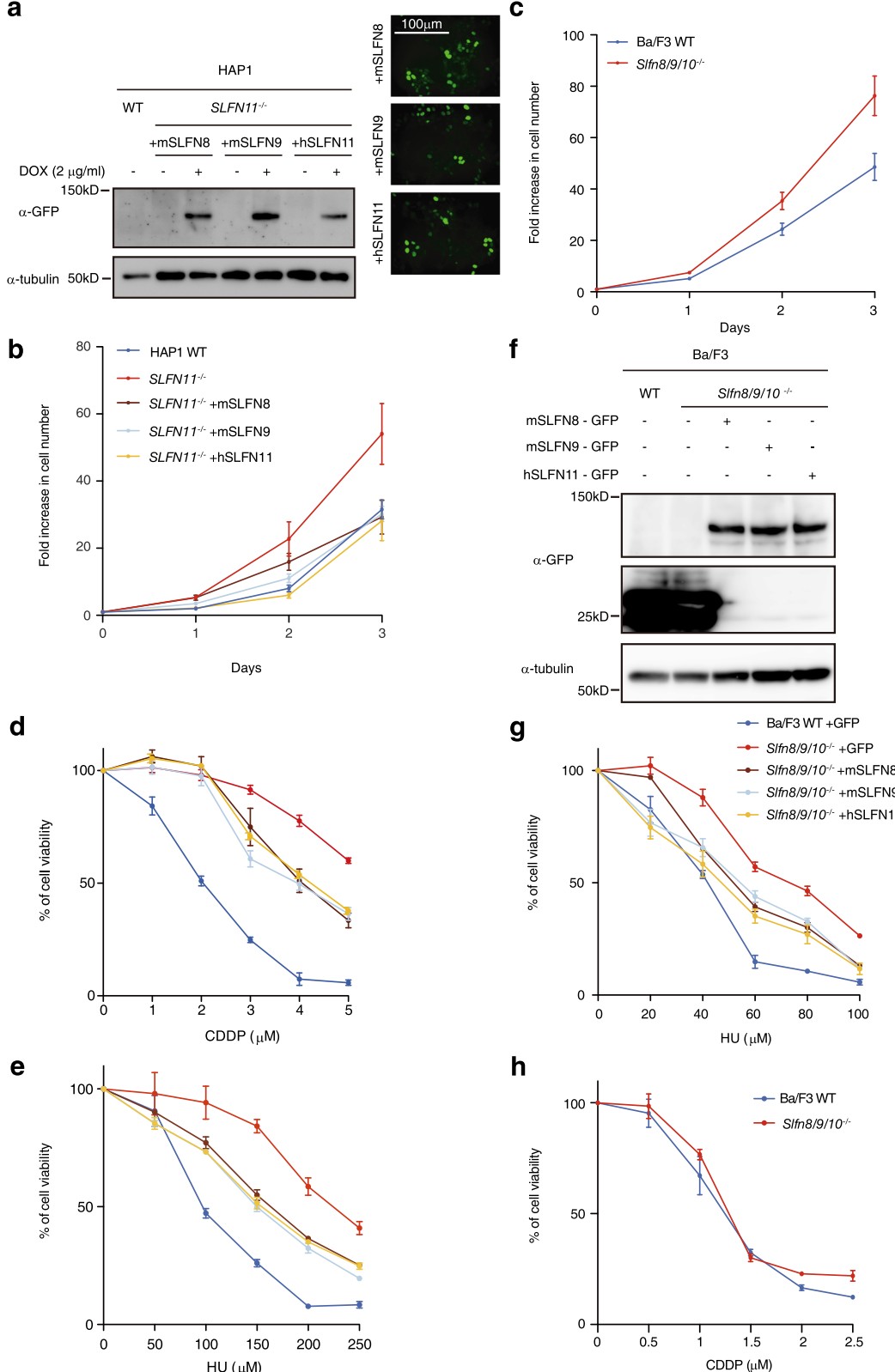

**Fig. 3 Analysis of *SLFNs* function in human and mouse cell lines. a** Western blotting (WB) and microscopic analysis of HAP1 *SLFN11*$^{-/-}$ cell line with DOX-induced expression of SLFNs-GFP. **b** Cell proliferation profile of the HAP1 cell lines with the indicated genotypes. **c** Cell proliferation profile of the Ba/F3 *Slfn8/9/10*$^{-/-}$ cell line. **d** CDDP or **e** HU sensitivity of the HAP1 cell lines with the indicated genotypes. **f** Western blotting of Ba/F3 *Slfn8/9/10*$^{-/-}$ cell line with transient expression of SLFNs-GFP, and **g** HU or **h** CDDP sensitivity of the *Slfn8/9/10*$^{-/-}$ cells with the indicated transgenes. Expression of GFP alone was included as a control. Mean ± SD in quadruplicate cultures is shown.

lead to chemoresistance in cancer cells[19,30]. Nucleases such as MRE11 and DNA2 degrade nascent DNA when stalled replication forks are reversed, and proteins such as RAD51 and BRCA2 counteract this degradation by protecting the fork. Individual replication forks and whether they are degraded can be visualized in a DNA fiber assay wherein DNA is pulse-labeled with IdU followed by CldU and treated with HU to stall the replication forks[31]. In this assay, chromatin is spread onto slides, fixed, and stained with fluorescent anti-IdU or CldU antibodies allowing the length of each replication tract to be measured (Fig. 4a). If treatment with HU degrades the replication fork, the length of CldU (stained green) fibers is shortened[31].

We have previously reported that *SLFN11* can prevent recruitment of the fork protector RAD51 to the nascent DNA strand and accelerate stalled fork degradation[19]. We have tested whether *Slfn8* and *Slfn9* expression in HAP1 *SLFN11⁻/⁻* cells can affect HU-induced fork degradation via the DNA fiber assay. Consistent with the previous findings[19], we could confirm that *SLFN11⁻/⁻* cells with and without HU treatment had no significant difference in CldU fiber length, while in WT cells the CldU tracts were shortened (degraded) following HU treatment. In *SLFN11⁻/⁻* cells with DOX-induced expression of *SLFN11, Slfn8,* or *Slfn9*, there was a significant shortening in the CldU tract length after HU treatment (Fig. 4b). However, the expression of *Slfn2* did not lead to a similar CldU tract shortening (Supplementary Fig. 7). Interestingly, SLFN11-K652D still exhibited some level of degradation after HU treatment (Supplementary Fig. 7). In Ba/F3 cell lines, we observed little degradation in both WT and *Slfn8/9/10⁻/⁻* cells when treated with HU. This could be due to the potentially limited expression levels of *Slfn8/9*, or perhaps more robust fork protection activities exist in Ba/F3. Consistent with this notion, the stalled fork was degraded after transient expression of *SLFN11, Slfn8,* or *Slfn9*, similar to HAP1 cell lines (Fig. 4c). Taken together, these results suggest that *Slfn8* or *Slfn9* expression complements the loss of *SLFN11* in human cells. Further, this expression allows destabilization of the stalled and reversed replication fork, and thus, fork degradation, in both human and mouse cells.

**SLFNs prevent RPA and RAD51 recruitment to DNA damage sites in HAP1 cells.** We have previously reported that RPA and RAD51 recruitment to DNA damage-induced foci or the nascent DNA strand at stalled replication forks was enhanced by *SLFN11* depletion[19]. To examine the effects of *Slfn8 or Slfn9* expression in *SLFN11⁻/⁻* cells, we tested the levels of HU-induced foci formation. As expected, both RPA and RAD51 foci levels were decreased after DOX-induced expression of *Sln8* or *Slfn9* as well as *SLFN11* in HAP1 cells (Fig. 5a, b). However, the expression of *Slfn2* did not show a similar reduction in the foci (Supplementary Fig. 8a, b). The foci levels in cells expressing SLFN11-K652D were similar to HAP1 *SLFN11⁻/⁻* cells, indicating that SLFN11 may prevent RPA and RAD51 recruitment depending on its ssDNA binding capability (Supplementary Fig. 8a, b). These results suggest that similarly to *SLFN11, Slfn 8* and *Slfn9* can prevent RPA and RAD51 recruitment to DNA damage sites, or by extension, to stalled forks in human cells.

## Discussion
The cross-species correspondence within subgroup III *SLFNs*, such as between human *SLFN11* and *SLFN13* versus mouse *Slfn8* and *Slfn9*, have remained elusive. In this study, we addressed this issue by asking if there is any functional similarity between human *SLFN5/11/13* and mouse *Slfn8/9*. We conclude that *Slfn8* and *Slfn9* share the function with human *SLFN11*, thus could be its orthologs from the following observations. First, we observed the similar

behavior of hSLFN11, but not mSLFN2, hSLFN5 or hSLFN13, to that of mSLFN8/9 in recruitment to the laser-induced DNA damage tracks. Second, we examined the phenotype of *Slfn8/9/10* knockout in the mouse B cell line Ba/F3, and observed similarities in cell growth and DNA damage sensitivity to the human *SLFN11⁻/⁻* cells. Third, we tested whether the expression of *SLFN11* or *SLFN8/9* could complement mouse or human knockout cell lines, respectively, across the species. We found that both *SLFN11* and *Slfn8/9* could similarly complement *SLFN11* loss in human cells or *Slfn8/9* loss in mouse cells. Fourth, the introduction of *Slfn8* and *Slfn9* into *SLFN11⁻/⁻* cells could destabilize HU-stalled replication forks, as shown by DNA fiber analysis, in a similar manner to *SLFN11*. Finally, we provide evidence that mSLFN8/9 can prevent RPA or RAD51 recruitment upon replication stress in human HAP1 cells, which could contribute to the increased fork destabilization and the DNA damage sensitivity we observed. These findings corroborate our previous report about the function of *SLFN11*[19]. However, the expression of K652D loss of ssDNA binding mutant in *SLFN11⁻/⁻* cells showed mild enhancement of HU sensitivity and fork degradation without affecting RPA/RAD51 foci formation. This may suggest presence of another mechanism for increased stalled fork degradation mediated by SLFN11 other than RAD51 regulation. Of note, recent studies have implicated increased levels of single-strand (ss) gaps, rather than resected stalled forks, in sensitizing cells to DNA damaging treatments such as PARP inhibitors[32]. It remains unclear whether *SLFN11* expression can affect levels of ssDNA gaps.

In our laser track experiments, the K652D mutation of hSLFN11 showed a definite impact on SLFN11 recruitment to the microlaser track, indicating that the ssDNA binding plays a key role in the acute phase of SLFN11 recruitment[8,16]. SLFN11 appeared to regulate RPA/RAD51 foci levels via ssDNA binding or this could be due to the recruitment defects. On the other hand, hSLFN11 accumulation seemed to partially depend on its interaction with the single strand binding protein RPA, and we also confirmed that RPA itself was rapidly recruited. These observations might be related to the recent report that Pol III is quickly recruited to DNA break sites and initiates RNA synthesis, leading to DNA: RNA hybrids that may protect the 3' overhang[33]. The displaced 5' end of the DNA strand may be bound by RPA as well as SLFN11. Interestingly, it has been reported that hSLFN11 interacts with DHX9 helicase[16], which may function to regulate R-loops[34,35]. Thus SLFN11 may be initially recruited to ssDNA created at the DNA damage sites, then the binding could be stabilized by interacting with RPA that are also recruited by ssDNA. However, it should be noted that our microlaser experiments chased SLFN11 accumulation only for 1–2 min. The kinetics of hSLFN11 recruitment and contribution of its ssDNA binding to the accumulation at damaged DNA ends should warrant further investigation.

In conclusion, our functional analyses supports the notion that mouse *Slfn8* and *Slfn9* can have functions similar to human *SLFN11*, and therefore we propose that they are the orthologs of *SLFN11* at least in some of the functional aspects. However, it is still possible that they share the biological role assigned to human *SLFN13*, even though the predominant subcellular localization of mSLFN8/9 and hSLFN13 may differ because of the presence or absence of the NLS. It is currently unclear why mice evolved to carry two copies of putative *SLFN11* homologs and to how much degree these two homologs have overlapping functions. To the best of our knowledge, the phenotype of the *Slfn8* single knockout mice has been described[36]. Given the possible redundancy between *Slfn8* and *Slfn9*, it would be worthwhile to characterize double knockout mice lacking both of these *Slfns*. Such mouse models might be useful in studying various conditions such as cancer development, chemotherapeutic responses, or Fanconi anemia.

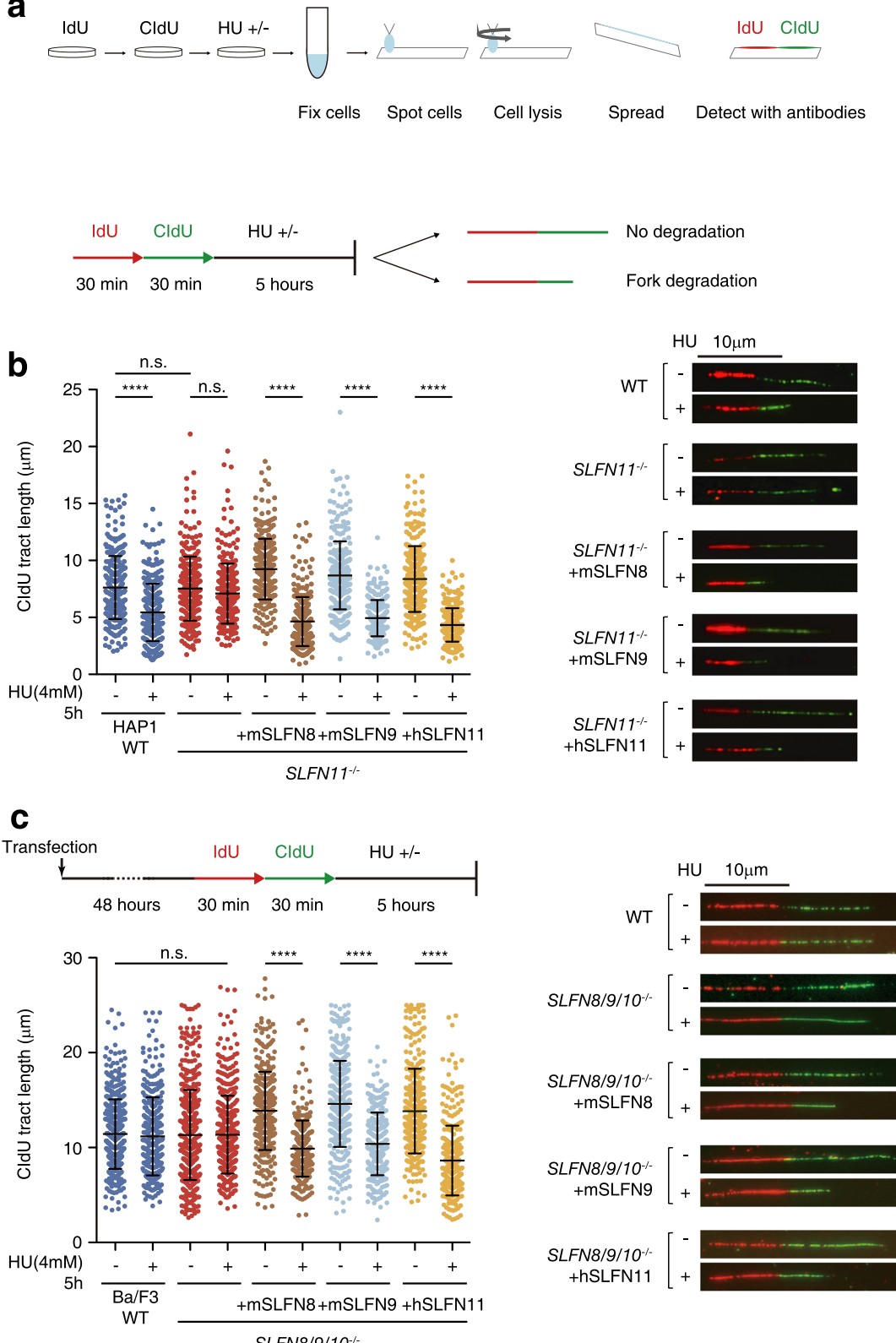

**Fig. 4 Recovery of replication fork degradation upon *SLFN11* or *Slfn8/9* complementation in *SLFN11*−/− cells. a** A schema of the DNA fiber assay protocol. **b** Mouse *Slfn8/9* and human *SLFN11* restore replication fork degradation after HU treatment in HAP1 *SLFN11*−/− cells. **c** Mouse *Slfn8/9* or human *SLFN11* expression enhanced the replication fork degradation after HU treatment in Ba/F3 *Slfn8/9/10*−/− cells. For each sample, the length of 300 CldU tracts was measured. The *P* values were calculated using one-way ANOVA with Tukey's multiple-comparisons test. To minimize observer bias, the images were captured and analyzed in a blinded manner. Represent images are shown. Mean ± SD (*n* ≥ 300) are shown. n.s. : not significant. **** : *p* < 0.0001.

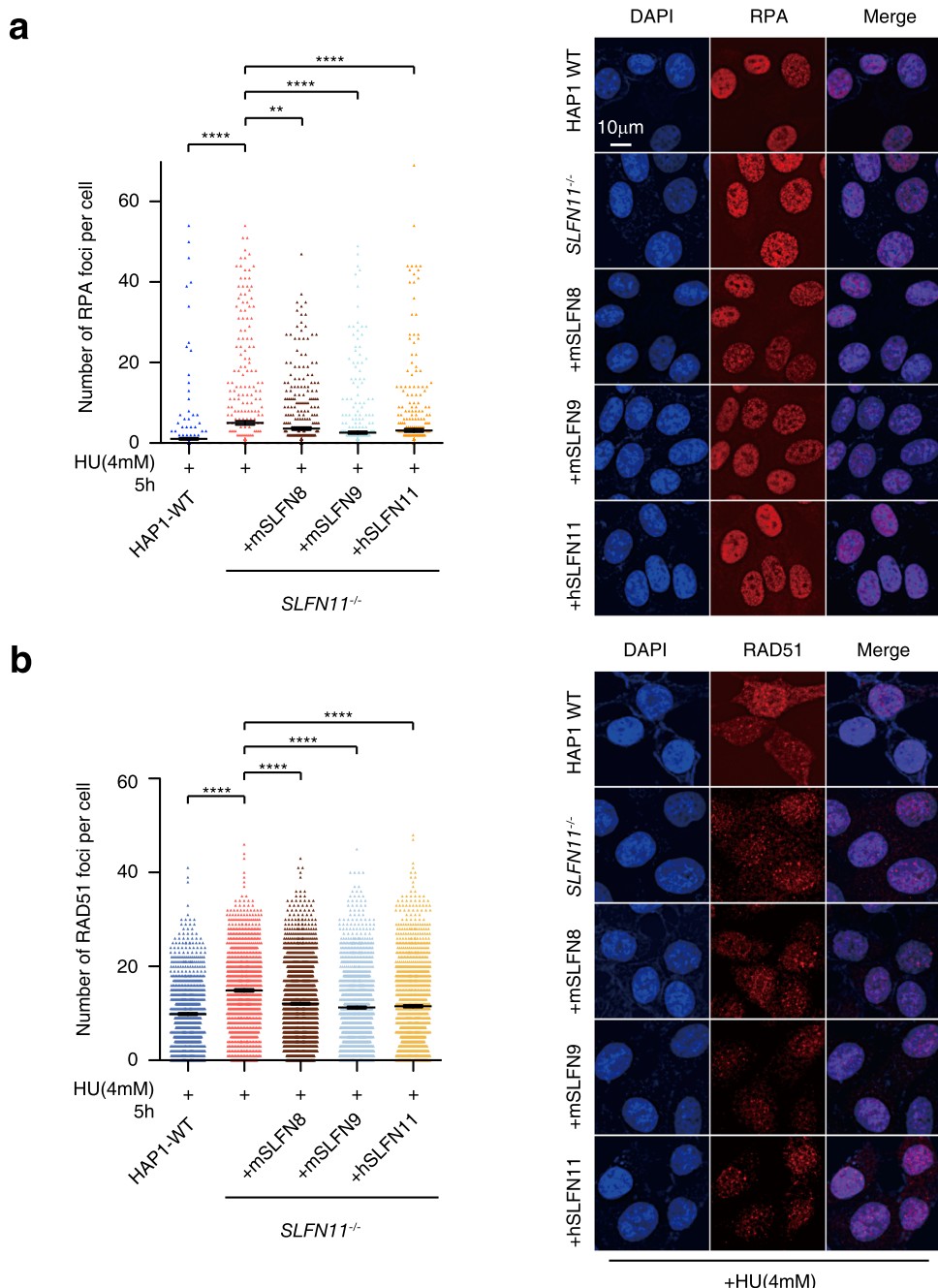

**Fig. 5 Decreased RAD51 and RPA foci levels upon *SLFN11* or *Sln8/9* complementation in *SLFN11*$^{-/-}$ cells.** Quantification of **a** RPA and **b** RAD51 foci per cell in HAP1 cell derivative with the indicated genotypes. Each dot represents the number of foci per nucleus in a single cell. Cells were exposed to HU 4 mM for 5 h and stained with the indicated antibodies. Mean ± SEM ($n \geq 500$) are shown for each condition. The experiment was repeated twice with similar results. The *P* values were calculated using one-way ANOVA with Tukey's multiple-comparisons test. Representative images are shown. n.s.: not significant. **$p < 0.01$. ****$p < 0.0001$.

## Methods

**Protein sequence alignments**. To analyze homology between the protein sequences of each SLFN of interest, NCBI protein sequences (hSLFN11: NP_001098057.1; hSLFN13: AAI36623.1; hSLFN5: NP_659412.3; hSLFN14: NP_001123292.1; mSLFN8: NP_853523.2; mSLFN9: NP_766384.2; mSLFN5: NP_899024.3; mSLFN14: NP_001159500.1) were used and MAFFT alignment using Genetyx software was performed (GENETYX Corp. Tokyo, Japan).

**Cell culture**. HAP1 cells were cultured in IMDM (Nacalai Tesque) with 10% Fetal Bovine Serum (FBS). HEK293T or U2OS cells were cultured in DMEM-high glucose (Nacalai Tesque) supplemented with 10% FBS. Ba/F3 cells were cultured in IMDM with 10% FBS and interleukin-3 (IL-3, 1 μg/mL, Biolegend). hTert1-RPE1 were maintained in DMEM:F12 supplemented with 10% FBS and hygromycin 0.01 mg/ml. HL60 were cultured in RPMI1640 supplemented with 10% FBS.

**Plasmid construction**. The full-length *Slfn8/9* (derived from Ba/F3), *SLFN5* (hTert1-RPE1), and *SLFN13* (HL60) cDNAs were isolated by PCR using the reverse transcription (RT) product each derived from the respective cell line as a template and cloned into

pENTR D-TOPO plasmid (Invitrogen). *Slfn2* cDNA was amplified from pEZT-GST-SLFN2 (Addgene plasmid # 174319). PrimeScript RT reagent Kit was used to carry out the RT reaction following the manufacturer's instructions. Mutants were generated using KOD One polymerase kit (TOYOBO) with the inverse PCR strategy and confirmed by Sanger sequencing. The coding sequences in pENTR were transferred into the CSIV lentiviral plasmid (RIKEN) or pcDNA3.1 (Invitrogen) using Gateway LR clonase II (Invitrogen). Human RPA1-GFP was previously described[37] and transferred into CSIV.

**Construction of *SLFN* knockout cell lines and exogenous expression of *SLFNs*.** The generation of *SLFN11*[−/−] HAP1 cells was described previously[19]. To generate Ba/F3 *Slfn*8/9/10 knockout cell line, the targeting vector was made from PCR-amplified genomic fragments and the resistance gene cassette using GeneArt seamless cloning and assembly enzyme mix (Invitrogen) as indicated in Supplementary Fig. 4. The CRISPR plasmid was made by inserting the annealed oligonucleotide containing a gRNA sequence targeting either *Slfn8* exon 3 or *Slfn9* exon 4 into the BbsI site of pX330 (Addgene #42230, a gift from Dr. Feng Zhang) using conventional T4 ligase cloning. The targeting vector and two CRISPR plasmids were transfected into Ba/F3 cells using Neon Transfection System 100 μL Kit (1600 V, 10 ms, 3 pulses), and selected with 1 μg/mL puromycin. Correctly edited cells were identified by PCR-mediated analysis of genomic DNA and confirmed by RT-PCR. To generate the lentivirus, HEK293T cells were transfected with CSIV plasmid, together with packaging constructs pCAG-HIVgp, and pCMV-VSV-G-RSV-rev using the Lipofectamine3000 reagent (Invitrogen) according to the manufacturer's instructions. After 48 h, the medium was carefully passed through a 0.22 μm filter and applied to HAP1 *SLFN11*[−/−] cells. Infected cells were selected with hygromycin 400 μg/mL (Nacalai Tesque). Single clones were isolated and verified by western blotting. GFP-tagged *SLFNs* cloned in pcDNA3.1 vector were transiently expressed in Ba/F3 cells by Neon as above.

**siRNA transfections.** The siRNA duplexes used in this study were purchased from Invitrogen. Transfection and co-transfection were carried out using Lipofectamine RNAiMAX (Invitrogen) according to the manufacturer's instructions. The siRNA duplexes used were: siRPA1 (5′rGrGrArAUUrAUrGrUrCrGUrArArGUrCrATT; 5′UrGrArCUUrArCrGrArCrAUrArAUUrCrCTT) (Sigma-Aldrich).

**Western blotting.** Samples were separated by SDS-PAGE (sodium dodecyl sulfate-polyacrylamide gel electrophoresis) and transferred to a polyvinylidene difluoride (PVDF) membrane and probed using indicated antibodies. The detection was done using ECL western blotting reagents (Sigma-Aldrich). The antibodies used in this study were listed in Supplementary Table 1.

**RT-PCR assay to determine mRNA expression of *SLFNs*.** Total RNA was isolated by RNeasy kit (Qiagen) and cDNA was synthesized by PrimeScript[TM] RT reagent kit with gDNA Eraser (TaKaRa). PCR amplification was carried out using KOD-FX polymerase. These experiments were carried out according to the manufacturer's instructions with a lower cycle number to avoid the plateau effects. The primers used in this study were listed in Supplementary Table 2.

**Cell growth assay and cytotoxicity assay.** HAP1 cells ($1 \times 10^5$) and Ba/F3 cells ($1 \times 10^5$) were seeded into 6 cm dishes at day 0 and counted every 24 h. For cytotoxicity assays, Ba/F3 cells ($2.5 \times 10^3$) or HAP1 cells ($2.5 \times 10^3$) were plated in a 96-well plate in quadruplicate for each condition. 48 h after DOX addition, the indicated amounts of HU or CDDP were added to the wells and incubated for a further 72 h. The HU or CDDP concentrations were chosen based on previous studies[19,38] and in the similar range to the plasma concentration with the clinical relevance[39,40]. Cell viability was assayed using an MTS reagent (Nacalai Tesque). Absorbance at 450 nm was measured with a Multilabel Reader (PerkinElmer).

**DNA fiber assay.** DNA fiber assay was carried out essentially as described before[19] but this time in a blinded manner. Cells undergoing exponential growth were incubated with 25 μM IdU for 30 min, then washed with PBS, and incubated with 250 μM CldU for an additional 30 min. Then they were incubated with or without 4 mM HU for 5 h before collection and suspension in 70% ethanol at a final concentration of $5.5 \times 10^5$ cells/mL. After spotting the cell suspension onto glass slides, cells were lysed with a solution of 50 mM EDTA, 0.5% SDS, and 200 mM Tris-HCl (pH7.5); and mixed using a circular motion with a pipette tip. Then the slides were tilted at 15° to spread the DNA fiber across the surface. After drying, fibers were fixed in a solution of methanol: acetic acid (3:1) in a staining jar, and then denatured with 2.5 M HCl for 60 min. Then the slides were washed in PBS 3 times. The slides are blocked using Blocking One (Nacalai Tesque) for 20 min. The primary antibodies used were anti-BrdU from BD (for IdU, mouse) and anti-BrdU from Abcam (for CldU, rat) diluted to 1:400 in Blocking One, and added to slides for one hour in a humidified chamber. The slides are washed with PBST 3 times before incubation with the secondary antibodies. The secondary antibodies are anti-mouse Alexa594 and anti-rat Alexa488 in a 1:500 dilution. Finally, slides are washed with PBST and PBS, then a mounting medium (Prolong gold antifade reagent, Invitrogen) is added and topped with cover glass, then sealed with nail polish to protect the slides. Fibers were measured using the Leica DM5500B microscope and Leica Application Suite X (LAS X) software.

**Microlaser irradiation experiments.** A Leica TCS/SP5 confocal microscopy equipped with the 405-nm diode laser system was used for irradiation. U2OS cells were transiently transfected with pcDNA3.1 constructs encoding GFP-tagged *SLFN* or infected with CSIV-RPA1-GFP lentivirus and kept in a 37 °C heated chamber with 5% $CO_2$ and treated with 10 μg/ml of Hoechst 33342 (Thermo Fisher Scientific) for 10 min. Living cells were visualized with a 63×/1.40 oil objective lens. DNA damage was induced by irradiation with a 405-nm diode laser. Leica LAS AF software was used for the acquisition of images.

**Immunohistochemistry.** HAP1 cells were fixed with 3% paraformaldehyde and 2% sucrose in PBS and then permeabilized with 0.1% Triton X-100/PBS for 10 min. After blocking with 2% BSA/PBS, slides were incubated with indicating antibodies, followed by incubation with secondary antibodies. Nuclei were counterstained with DAPI (Sigma-Aldrich). The number of foci was enumerated using an INCellAnalyzer2000 instrument (Cytiva).

**Statistics and reproducibility.** In the experiment of microlaser-induced DNA damage sites, more than 10 irradiated U2OS cells are shown, and the experiment was repeated more than 3 times. For the HU or CDDP treated sensitive assay, the experiments were performed in quadruplicate cultures and repeated twice. The DNA fiber assay was performed in a blamed manner, and reproduced from different individuals. The foci levels of RPA and Rad51 were analyzed in more than 500 cells by the INCellAnalyzer2000 instrument (Cytiva), and the experiment was repeated twice with similar results. One representative set of data is shown. The

*P* values were calculated using one-way ANOVA with Tukey's multiple-comparisons test in Prism software (Graphpad, USA).

## Data availability

All relevant data are available from the authors upon reasonable request. Uncropped and unedited blot images for all figures are provided in Supplementary Figs. 9, 10, 11,12, and 13. Source data for all graphs in the manuscript are provided in the Supplementary Data 1.

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

## Acknowledgements

We would like to thank Drs. Junko Murai, Yasuhisa Murai, and Kiichiro Tsuchiya, for discussions; Dr. Andres Canela for critical reading of the manuscript and discussion; Dr. Feng Zhang for pX330; Dr. Bruce Beutler for mSLFN2 plasmid; Dr. Hitoshi Kurumizaka for anti-RAD51 serum; the late Dr. Hiroyuki Miyoshi and RIKEN BRC for the Lentivirus system; Dr. Koichi Sato for advice for Alfafold 2 structural prediction; Ms. Masami Tanaka, Mayu Yamabe, Sumiko Matsui, Xuye Wang, and Lin Liu for technical and secretarial assistance. Anfeng Mu is supported by the Kyoto University Research Coordination Alliance. This work is also partly supported by the KAKENHI Kiban B (Grant# 20H03450 to M.T.), Takeda Science Foundation (to A.M.), The Uehara Memorial Foundation (to A.M.), and JSPS Core-to-Core Program (Grant# JPJSCCA20200009).

## Author contributions

A.M., Y.O. and M.T. designed the study. E.A. compared the protein sequences, cloned the cDNAs and carried out DNA fiber analysis with a help from M.O. and F.Q. A.L.M. made Ba/F3 *Slfn8/9/10* knockout cell lines. Y.K. performed laser track experiments. E.A., M.T. and A.M. wrote the manuscript.

## Competing interests

The authors declare no competing interests.

## Additional information

**Peer review information** : *Communications Biology* thanks the anonymous reviewers for their contribution to the peer review of this work. Primary Handling Editors: Valeria Naim and George Inglis. A peer review file is available.

