## [Peer Review File · Communications Biology]

Reviewers' comments:

Reviewer #1 (Remarks to the Author):

The manuscript by Alvi et al. presents the identification of the mouse orthologue of human SLFN11. The authors compared the function of mouse SLFN8 and SLFN9 with that of hSLFN11. Additionally, they attempted to complement the loss of human SLFN11 in SLFN11^{-/-} cells with mSLFN8 or mSLFN9. Recently, SLFN11 has gained attention for its role as a viral restriction factor and biomarker in certain cancer types. In particular, the correlation between the expression of SLFN11 and cell death of cancer cells following DNA-damaging cancer chemotherapies has led to increased research interest. This study could facilitate further *in vivo* investigations in mice, which could help elucidate the biological role of SLFN11 in greater detail.

Unfortunately, the authors have overlooked the structure of human SLFN11 and hence did not discuss the results in the light of this data. In particular, the structure bound to ssDNA could presumably give some hints to the orthologous schlafen family member in mice. Are the residues involved in single-strand DNA binding and dimerization of SLFN11 conserved in mouse SLFN8/9?

Although AlphaFold 2 was a bit misleading in predicting the SLFN11 structure a couple of months ago, I would recommend predicting mSLFN8/9 with AlphaFold 2, comparing it with hSLFN5 and hSLFN11, and discussing the results concerning the active site and substrate recognition residues.

Actually, the SWADL domain is in reality a SWAVDL domain.

The authors and others report that SLFN11 is recruited to stalled replication forks via direct interaction with RPA. However, in Figure 5a, it is shown that in SLFN11^{-/-} cells, the number of RPA foci is increased and by addition of mSLFN8/9 or hSLFN11, the number of RPA foci is reduced again after HU treatment. I am not an expert in this type of assay, but if SLFN11 is recruited via RPA, why would the addition of SLFN11/mSLFN8/mSLFN9 remove RPA from stalled replication forks? Additionally, I am not convinced by the statistical significance in Figure 5a and 5b. I would expect a more significant influence on one hand in the knockout and on the other hand in the overexpression setting. Along the same line, in the RPA knockdown in Fig. 2a, SLFN11 still localizes to microlaser-induced DNA damage sites, which in my opinion also speaks against direct recruitment of SLFN11 by RPA. A binding to single-stranded DNA at the blocked replication fork is much more intuitive.

In summary, I think the results of this work are important for the field to facilitate functional studies in mouse models. However, a more thorough discussion of the current literature is necessary. The RPA colocalization versus replacement should be analyzed in more detail or better described. A deletion of the NLS as in the SLFN11 $\Delta 5$ would ultimately lead to exclusion from the nucleus, and a better-designed experiment (e.g. point mutations in the ssDNA binding region) would be more suitable to clarify the direct interaction with RPA versus binding to ssDNA at stalled replication forks.

Reviewer #2 (Remarks to the Author):

This is an interesting manuscript in which the authors demonstrate that mouse Slfn8/9 expression can complement human SLFN11 loss in human SLFN11^{-/-} cells, reducing the growth rate to wild-type levels and partially restoring sensitivity to DNA-damaging agents. The authors also found that both Slfn8/9 and SLFN11 expression accelerate stalled fork degradation and decrease RPA and RAD51 foci after DNA damage. The authors therefore propose that mouse Slfn8 and Slfn9 genes may share an orthologous function with human SLFN11. These findings are novel, interesting and of importance in the field. In general, the work is carefully done and provides important new information. However, there are some specific issues outlined below that require attention.

1. Although the selection of Slfn8/Slfn9 to complement SLFN11^{-/-} cells was well thought, it would be important to have some additional negative controls for the studies of figures 3, 4 and 5. An additional mouse Slfn gene not predicted to complement could be used as negative control for the effects of the complementation of Slfns 8 and 9. For instance, the authors could use mouse Slfn2 or Slfn5 to

complement SLFN11^{-/-} cells as controls.

2. In Figure 3, panel b, the doses of hydroxyurea used are very high, raising the question of specificity and biological relevance of these in vitro observations. The authors should at least comment and discuss this.

Dear Dr Naim,

According to your suggestion, we have mainly addressed two points: (1) SLFN11 ssDNA/RPA binding, and (2) the inclusion of mSLFN2 as a requested negative control. These are explained below in a point-by-point manner in response to the reviewer's comments. While the mechanistic elucidation of the SLFN11 function is very interesting and we are still working on these aspects, we would like to maintain our focus in this manuscript as an identification of possible SLFN11 homologs in mice.

Reviewers' comments:

Reviewer #1 (Remarks to the Author):

The manuscript by Alvi et al. presents the identification of the mouse orthologue of human SLFN11. The authors compared the function of mouse SLFN8 and SLFN9 with that of hSLFN11. Additionally, they attempted to complement the loss of human SLFN11 in SLFN11^{-/-} cells with mSLFN8 or mSLFN9. Recently, SLFN11 has gained attention for its role as a viral restriction factor and biomarker in certain cancer types. In particular, the correlation between the expression of SLFN11 and cell death of cancer cells following DNA-damaging cancer chemotherapies has led to increased research interest. This study could facilitate further *in vivo* investigations in mice, which could help elucidate the biological role of SLFN11 in greater detail.

Unfortunately, the authors have overlooked the structure of human SLFN11 and hence did not discuss the results in the light of this data. In particular, the structure bound to ssDNA could presumably give some hints to the orthologous schlafen family member in mice. Are the residues involved in single-strand DNA binding and dimerization of SLFN11 conserved in mouse SLFN8/9?

Thank you for bringing up the structure of SLFN11 and its relevance to our study. We have noticed the paper on SLFN11 structure (Metzner et al. Nat Comm 2022) but failed to discuss about it in our original manuscript. Now the findings in the paper are cited and discussed (p.3, line 16). Regarding structural conservation, we have found that the SLFN11 ssDNA binding site is conserved in mouse SLFN8/9 (in SLFN5 and 13 as well), indicating a likely functional similarity in their ssDNA binding capabilities. Additionally, the dimer interfaces are well conserved, although there are some variations in the dimer interface II residues between SLFN11 and SLFN8/9. However, we think this does not exclude the possibility of similar dimerization in SLFN8 or SLFN9. We added the discussion on p.6 line 14 and also indicated the conserved amino acid residues across the ssDNA binding domain and dimerization domain by marking them in Supplementary Figure 2.

Although AlphaFold 2 was a bit misleading in predicting the SLFN11 structure a couple of months ago, I would recommend predicting mSLFN8/9 with AlphaFold 2, comparing it with hSLFN5 and hSLFN11, and discussing the results concerning the active site and substrate recognition residues.

Thank you for your valuable suggestion regarding predicting mSLFN8/9 structure with AlphaFold 2 and comparing it with hSLFN5 and hSLFN11. We looked at the predicted structures of mSLFN5/8/9/11/13 with AlphaFold 2 (<https://alphafold.ebi.ac.uk/>). Since we are not experts in structural analysis, we consulted our collaborator and structural biologist Dr. Koichi Sato –Hubrecht Institute, who is now included in the Acknowledgement. We concluded that the predicted model structures appear quite similar and we thus couldn't determine which SLFN family members are in an orthologous relationship with human SLFN11 based on the prediction. In the revised manuscript, we emphasised this overall similarity in the predicted structures to highlight the importance of functional analysis (p.6, line 17).

Actually, the SWADL domain is in reality a SWAVDL domain.

Thank you for bringing this to our attention. We have corrected the name.

The authors and others report that SLFN11 is recruited to stalled replication forks via direct interaction with RPA. However, in Figure 5a, it is shown that in SLFN11^{-/-} cells, the number of RPA foci is increased and by addition of mSLFN8/9 or hSLFN11, the number of RPA foci is reduced again after HU treatment. I am not an expert in this type of assay, but if SLFN11 is recruited via RPA, why would the addition of SLFN11/mSLFN8/mSLFN9 remove RPA from stalled replication forks?

Thank you for raising this issue. According to the reviewer's very interesting suggestion, we examined the effects of K652D mutation (loss of ssDNA binding) on SLFN11 recruitment (Figure 2b). As we discussed in the revised manuscript (p.7, line 12), we modified our previous view and now conclude that, in very early phase following DNA damage (1-2 min), the ssDNA binding plays a key role in SLFN11 recruitment, while RPA binding contributes only partially. We hope the reviewer agrees with our interpretation of the data. Given the remaining SLFN11 function in the K652D mutant (enhanced fork degradation and HU sensitivity), it seems possible that SLFN11 may relocalize to DNA damage/replication stress sites at later time points in a manner independent of ssDNA binding. In any case, further research is needed to elucidate the precise molecular mechanisms involved in the interplay between SLFNs and RPA at stalled replication forks.

Additionally, I am not convinced by the statistical significance in Figure 5a and 5b. I would expect a more significant influence on one hand in the knockout and on the other hand in the overexpression setting.

Thank you for raising your concern regarding the statistical significance in Figures 5a and 5b. We have carried out those experiments several times, and the results consistently showed very similar trends as expected from previous data (Okamoto et al., Qi et al., and other researchers' work). To make the figures clearer for readers, we modified Figure 5 by removing data from cells without HU treatment. We hope the changes made the figures more convincing. Please note the strong statistical significance as shown by the p-values.

Along the same line, in the RPA knockdown in Fig. 2a, SLFN11 still localizes to microlaser-induced DNA damage sites, which in my opinion also speaks against direct recruitment of SLFN11 by RPA. A binding to single-stranded DNA at the blocked replication fork is much more intuitive.

A deletion of the NLS as in the SLFN11 $\Delta 5$ would ultimately lead to exclusion from the nucleus, and a better-designed experiment (e.g. point mutations in the ssDNA binding region) would be more suitable to clarify the direct interaction with RPA versus binding to ssDNA at stalled replication forks.

We agree. As described above, we carried out the microlaser experiments using the K652D mutant and the results were as expected by the reviewer. We changed our interpretation and now described the role of RPA in SLFN11 recruitment only as a partial contribution (described in p.7, line 14). We also added more discussion (p.12, line 24-27, in Discussion).

Interestingly, SLFN11-K652D expression did not affect the HU-induced RPA or RAD51 foci levels, raising the possibility that SLFN11 promotes the dissociation of RPA through the ssDNA binding domain. The exact mechanisms underlying the interaction between SLFN11 and RPA/RAD51 remain unclear and require further exploration.

In summary, I think the results of this work are important for the field to facilitate functional studies in mouse models. However, a more thorough discussion of the current literature is necessary. The RPA colocalization versus replacement should be analyzed in more detail or better described.

Thank you for your valuable feedback and insightful comments. We appreciate your recognition of the importance of our research findings in advancing the field and facilitating functional studies in

mouse models. As described above, we have performed additional experiments, and the data and discussion are included in this revision. We hope the reviewer thinks this revision is satisfactory.

Reviewer #2 (Remarks to the Author):

This is an interesting manuscript in which the authors demonstrate that mouse Slfn8/9 expression can complement human SLFN11 loss in human SLFN11^{-/-} cells, reducing the growth rate to wild-type levels and partially restoring sensitivity to DNA-damaging agents. The authors also found that both Slfn8/9 and SLFN11 expression accelerate stalled fork degradation and decrease RPA and RAD51 foci after DNA damage. The authors therefore propose that mouse Slfn8 and Slfn9 genes may share an orthologous function with human SLFN11. These findings are novel, interesting and of importance in the field. In general, the work is carefully done and provides important new information. However, there are some specific issues outlined below that require attention.

Thank you for recognizing the potential importance of our manuscript.

1. Although the selection of Slfn8/Slfn9 to complement SLFN11^{-/-} cells was well thought, it would be important to have some additional negative controls for the studies of figures 3, 4 and 5. An additional mouse Slfn gene not predicted to complement could be used as negative control for the effects of the complementation of Slfns 8 and 9. For instance, the authors could use mouse Slfn2 or Slfn5 to complement SLFN11^{-/-} cells as controls.

Thank you for your valuable suggestion regarding the need for additional negative controls in our studies. Loss of SLFN5 has been shown to impair non-homologous end joining (NHEJ) and lead to genomic instability (Huang J et al., Mol Cell 2023). To address the reviewer's concern, we thus included mSlfn2 data as an additional negative control in microlaser irradiation experiments, hydroxyurea sensitivity, DNA fiber assay, and foci levels of RPA and RAD51.

2. In Figure 3, panel b, the doses of hydroxyurea used are very high, raising the question of specificity and biological relevance of these in vitro observations. The authors should at least comment and discuss this.

Thank you for bringing up the concern regarding the doses of hydroxyurea. The high doses of hydroxyurea used in this experiment were chosen based on previous studies, where similar concentrations have been employed to observe differential cell killing in cells with different

genotypes. Furthermore, the concentrations we used were in a similar range to the plasma concentration achieved in the clinic (for example, please see Marahatta and Ware. *Blood Cells Molecules and Diseases* 2017; Rajkumar et al. *J Clin Diagn Res* 2016). In administered humans, the peak plasma concentration of HU or cisplatin is described as $\sim 26 \pm 7 \mu\text{g/mL}$ ($\sim 330 \mu\text{M}$) or $5.37 \pm 1.47 \mu\text{g/mL}$ ($\sim 18 \mu\text{M}$), respectively. We included a comment in the Methods section, on p.15 line 23.

REVIEWERS' COMMENTS:

Reviewer #1 (Remarks to the Author):

The revised version of the manuscript has addressed all of the raised questions. Considering that the primary focus of this manuscript is the identification of potential SLFN11 homologs in mice, I recommend its publication.

Reviewer #2 (Remarks to the Author):

The authors have addressed satisfactorily all the important issues raised in the original review.

Dear Dr Naim,

According to your suggestion, we have mainly addressed two points: (1) SLFN11 ssDNA/RPA binding, and (2) the inclusion of mSLFN2 as a requested negative control. These are explained below in a point-by-point manner in response to the reviewer's comments. While the mechanistic elucidation of the SLFN11 function is very interesting and we are still working on these aspects, we would like to maintain our focus in this manuscript as an identification of possible SLFN11 homologs in mice.

Reviewers' comments:

Reviewer #1 (Remarks to the Author):

The manuscript by Alvi et al. presents the identification of the mouse orthologue of human SLFN11. The authors compared the function of mouse SLFN8 and SLFN9 with that of hSLFN11. Additionally, they attempted to complement the loss of human SLFN11 in SLFN11^{-/-} cells with mSLFN8 or mSLFN9. Recently, SLFN11 has gained attention for its role as a viral restriction factor and biomarker in certain cancer types. In particular, the correlation between the expression of SLFN11 and cell death of cancer cells following DNA-damaging cancer chemotherapies has led to increased research interest. This study could facilitate further *in vivo* investigations in mice, which could help elucidate the biological role of SLFN11 in greater detail.

Unfortunately, the authors have overlooked the structure of human SLFN11 and hence did not discuss the results in the light of this data. In particular, the structure bound to ssDNA could presumably give some hints to the orthologous schlafen family member in mice. Are the residues involved in single-strand DNA binding and dimerization of SLFN11 conserved in mouse SLFN8/9?

Thank you for bringing up the structure of SLFN11 and its relevance to our study. We have noticed the paper on SLFN11 structure (Metzner et al. Nat Comm 2022) but failed to discuss about it in our original manuscript. Now the findings in the paper are cited and discussed (p.3, line 16). Regarding structural conservation, we have found that the SLFN11 ssDNA binding site is conserved in mouse SLFN8/9 (in SLFN5 and 13 as well), indicating a likely functional similarity in their ssDNA binding capabilities. Additionally, the dimer interfaces are well conserved, although there are some variations in the dimer interface II residues between SLFN11 and SLFN8/9. However, we think this does not exclude the possibility of similar dimerization in SLFN8 or SLFN9. We added the discussion on p.6 line 14 and also indicated the conserved amino acid residues across the ssDNA binding domain and dimerization domain by marking them in Supplementary Figure 2.

Although AlphaFold 2 was a bit misleading in predicting the SLFN11 structure a couple of months ago, I would recommend predicting mSLFN8/9 with AlphaFold 2, comparing it with hSLFN5 and hSLFN11, and discussing the results concerning the active site and substrate recognition residues.

Thank you for your valuable suggestion regarding predicting mSLFN8/9 structure with AlphaFold 2 and comparing it with hSLFN5 and hSLFN11. We looked at the predicted structures of mSLFN5/8/9/11/13 with AlphaFold 2 (<https://alphafold.ebi.ac.uk/>). Since we are not experts in structural analysis, we consulted our collaborator and structural biologist Dr. Koichi Sato –Hubrecht Institute, who is now included in the Acknowledgement. We concluded that the predicted model structures appear quite similar and we thus couldn't determine which SLFN family members are in an orthologous relationship with human SLFN11 based on the prediction. In the revised manuscript, we emphasised this overall similarity in the predicted structures to highlight the importance of functional analysis (p.6, line 17).

Actually, the SWADL domain is in reality a SWAVDL domain.

Thank you for bringing this to our attention. We have corrected the name.

The authors and others report that SLFN11 is recruited to stalled replication forks via direct interaction with RPA. However, in Figure 5a, it is shown that in SLFN11^{-/-} cells, the number of RPA foci is increased and by addition of mSLFN8/9 or hSLFN11, the number of RPA foci is reduced again after HU treatment. I am not an expert in this type of assay, but if SLFN11 is recruited via RPA, why would the addition of SLFN11/mSLFN8/mSLFN9 remove RPA from stalled replication forks?

Thank you for raising this issue. According to the reviewer's very interesting suggestion, we examined the effects of K652D mutation (loss of ssDNA binding) on SLFN11 recruitment (Figure 2b). As we discussed in the revised manuscript (p.7, line 12), we modified our previous view and now conclude that, in very early phase following DNA damage (1-2 min), the ssDNA binding plays a key role in SLFN11 recruitment, while RPA binding contributes only partially. We hope the reviewer agrees with our interpretation of the data. Given the remaining SLFN11 function in the K652D mutant (enhanced fork degradation and HU sensitivity), it seems possible that SLFN11 may relocalize to DNA damage/replication stress sites at later time points in a manner independent of ssDNA binding. In any case, further research is needed to elucidate the precise molecular mechanisms involved in the interplay between SLFNs and RPA at stalled replication forks.

Additionally, I am not convinced by the statistical significance in Figure 5a and 5b. I would expect a more significant influence on one hand in the knockout and on the other hand in the overexpression setting.

Thank you for raising your concern regarding the statistical significance in Figures 5a and 5b. We have carried out those experiments several times, and the results consistently showed very similar trends as expected from previous data (Okamoto et al., Qi et al., and other researchers' work). To make the figures clearer for readers, we modified Figure 5 by removing data from cells without HU treatment. We hope the changes made the figures more convincing. Please note the strong statistical significance as shown by the p-values.

Along the same line, in the RPA knockdown in Fig. 2a, SLFN11 still localizes to microlaser-induced DNA damage sites, which in my opinion also speaks against direct recruitment of SLFN11 by RPA. A binding to single-stranded DNA at the blocked replication fork is much more intuitive.

A deletion of the NLS as in the SLFN11 $\Delta 5$ would ultimately lead to exclusion from the nucleus, and a better-designed experiment (e.g. point mutations in the ssDNA binding region) would be more suitable to clarify the direct interaction with RPA versus binding to ssDNA at stalled replication forks.

We agree. As described above, we carried out the microlaser experiments using the K652D mutant and the results were as expected by the reviewer. We changed our interpretation and now described the role of RPA in SLFN11 recruitment only as a partial contribution (described in p.7, line 14). We also added more discussion (p.12, line 24-27, in Discussion).

Interestingly, SLFN11-K652D expression did not affect the HU-induced RPA or RAD51 foci levels, raising the possibility that SLFN11 promotes the dissociation of RPA through the ssDNA binding domain. The exact mechanisms underlying the interaction between SLFN11 and RPA/RAD51 remain unclear and require further exploration.

In summary, I think the results of this work are important for the field to facilitate functional studies in mouse models. However, a more thorough discussion of the current literature is necessary. The RPA colocalization versus replacement should be analyzed in more detail or better described.

Thank you for your valuable feedback and insightful comments. We appreciate your recognition of the importance of our research findings in advancing the field and facilitating functional studies in

mouse models. As described above, we have performed additional experiments, and the data and discussion are included in this revision. We hope the reviewer thinks this revision is satisfactory.

Reviewer #2 (Remarks to the Author):

This is an interesting manuscript in which the authors demonstrate that mouse Slfn8/9 expression can complement human SLFN11 loss in human SLFN11^{-/-} cells, reducing the growth rate to wild-type levels and partially restoring sensitivity to DNA-damaging agents. The authors also found that both Slfn8/9 and SLFN11 expression accelerate stalled fork degradation and decrease RPA and RAD51 foci after DNA damage. The authors therefore propose that mouse Slfn8 and Slfn9 genes may share an orthologous function with human SLFN11. These findings are novel, interesting and of importance in the field. In general, the work is carefully done and provides important new information. However, there are some specific issues outlined below that require attention.

Thank you for recognizing the potential importance of our manuscript.

1. Although the selection of Slfn8/Slfn9 to complement SLFN11^{-/-} cells was well thought, it would be important to have some additional negative controls for the studies of figures 3, 4 and 5. An additional mouse Slfn gene not predicted to complement could be used as negative control for the effects of the complementation of Slfns 8 and 9. For instance, the authors could use mouse Slfn2 or Slfn5 to complement SLFN11^{-/-} cells as controls.

Thank you for your valuable suggestion regarding the need for additional negative controls in our studies. Loss of SLFN5 has been shown to impair non-homologous end joining (NHEJ) and lead to genomic instability (Huang J et al., Mol Cell 2023). To address the reviewer's concern, we thus included mSlfn2 data as an additional negative control in microlaser irradiation experiments, hydroxyurea sensitivity, DNA fiber assay, and foci levels of RPA and RAD51.

2. In Figure 3, panel b, the doses of hydroxyurea used are very high, raising the question of specificity and biological relevance of these in vitro observations. The authors should at least comment and discuss this.

Thank you for bringing up the concern regarding the doses of hydroxyurea. The high doses of hydroxyurea used in this experiment were chosen based on previous studies, where similar concentrations have been employed to observe differential cell killing in cells with different

genotypes. Furthermore, the concentrations we used were in a similar range to the plasma concentration achieved in the clinic (for example, please see Marahatta and Ware. *Blood Cells Molecules and Diseases* 2017; Rajkumar et al. *J Clin Diagn Res* 2016). In administered humans, the peak plasma concentration of HU or cisplatin is described as $\sim 26 \pm 7 \mu\text{g/mL}$ ($\sim 330 \mu\text{M}$) or $5.37 \pm 1.47 \mu\text{g/mL}$ ($\sim 18 \mu\text{M}$), respectively. We included a comment in the Methods section, on p.15 line 23.